# Rapid Detection of Predominant SARS-CoV-2 Variants Using Multiplex High-Resolution Melting Analysis

Liying Sun,[a,b] Liqin Wang,[a,b] Chi Zhang,[a,b] Yan Xiao,[a,b,c] Lulu Zhang,[a,b] Ziyuan Zhao,[a,b] Ⓓ Lili Ren,[a,b,c] Ⓓ Junping Peng[a,b]

[a]NHC Key Laboratory of Systems Biology of Pathogens, Institute of Pathogen Biology, Chinese Academy of Medical Sciences and Peking Union Medical College, Beijing, China

[b]Key Laboratory of Respiratory Disease Pathogenomics, Chinese Academy of Medical Sciences and Peking Union Medical College, Beijing, China

[c]Christophe Merieux Laboratory, Institute of Pathogen Biology, Chinese Academy of Medical Sciences and Peking Union Medical College, Beijing, China

Liying Sun and Liqin Wang contributed equally in this study. The order was determined by the corresponding author after negotiation.

**ABSTRACT** Coronavirus disease 2019, caused by the severe acute respiratory syndrome coronavirus 2 (SARS-CoV-2), poses a considerable threat to global public health. This study developed and evaluated a rapid, low-cost, expandable, and sequencing-free high-resolution melting (HRM) assay for the direct detection of SARS-CoV-2 variants. A panel of 64 common bacterial and viral pathogens that can cause respiratory tract infections was employed to evaluate our method's specificity. Serial dilutions of viral isolates determined the sensitivity of the method. Finally, the assay's clinical performance was assessed using 324 clinical samples with potential SARS-CoV-2 infection. Multiplex HRM analysis accurately identified SARS-CoV-2 (as confirmed with parallel reverse transcription-quantitative PCR [qRT-PCR] tests), differentiating between mutations at each marker site within approximately 2 h. For each target, the limit of detection (LOD) was lower than 10 copies/reaction (the LOD of N, G142D, R158G, Y505H, V213G, G446S, S413R, F486V, and S704L was 7.38, 9.72, 9.96, 9.96, 9.50, 7.80, 9.33, 8.25, and 8.25 copies/reaction, respectively). No cross-reactivity occurred with organisms of the specificity testing panel. In terms of variant detection, our results had a 97.9% (47/48) rate of agreement with standard Sanger sequencing. The multiplex HRM assay therefore offers a rapid and simple procedure for detecting SARS-CoV-2 variants.

**IMPORTANCE** In the face of the current severe situation of increasing SARS-CoV-2 variants, we developed an upgraded multiplex HRM method for the predominant SARS-CoV-2 variants based on our original research. This method not only could identify the variants but also could be utilized in subsequent detection of novel variants since the assay has great performance in terms of flexibility. In summary, the upgraded multiplex HRM assay is a rapid, reliable, and economical detection method, which could better screen prevalent virus strains, monitor the epidemic situation, and help to develop measures for the prevention and control of SARS-CoV-2.

**KEYWORDS** COVID-19, SARS-CoV-2, variant detection, high-resolution melting analysis, real-time PCR

Address correspondence to Junping Peng, pengjp@hotmail.com, or Lili Ren, renliliipb@163.com.

The authors declare no conflict of interest.

The 2019 novel coronavirus (severe acute respiratory syndrome coronavirus 2 [SARS-CoV-2]) gave rise to an outbreak of viral pneumonia in December 2019. According to the World Health Organization (WHO), there have been more than 600 million confirmed coronavirus disease (COVID-19) cases and over six million deaths worldwide (https://covid19.who.int). Coronaviruses are a family of enveloped, single-stranded, positive-strand RNA viruses that cause genomic instability during viral replication and thus are highly susceptible to mutations. During replication, the viral strains evolve through the generation of new mutations. After the first case of COVID-19 was

reported, the virus spread rapidly all over the world. Following mutations in the virus genome, new variants emerged. To date, the WHO defines five variants of concern (VOCs), Alpha, Beta, Gamma, Delta, and Omicron, which have all spread swiftly on a global scale (1–4). VOCs of SARS-CoV-2 can cause serious diseases, increase infectivity, reduce any viral neutralization via antibodies as elicited by prior infections or vaccinations, and limit the efficacy of vaccine immunity. The Delta variant was first detected in India on 1 October 2020 and was the most rapidly expanding variant with the widest range at that time, initiating the second and third pandemic waves during the second half of 2020. Compared to the other VOCs, Delta has high community transmissibility (5, 6). Its viral load is approximately 1,000-fold higher than that of the wild-type viral strain, which led to significant morbidity. The ongoing COVID-19 outbreak poses an extraordinary threat to global public health as new and more virulent strains of these viruses are continually emerging and expanding their geographic range (3). The Omicron variant was first identified in South Africa and Botswana, being reported to the WHO on 24 November 2021, as a novel variant. The WHO confirmed Omicron to be a VOC, based on its rapid transmission and the disease severity caused by this virus. Currently, the Omicron variant has been detected in over 60 countries and has become a predominant strain, second to only the Delta variant. Omicron has more rapid transmission and mutation sites than other VOCs (7), displaying over 30 mutations in the gene coding for the coronavirus spike (S) protein and some missense sites. Among them, 15 mutations are in the receptor binding domain (RBD), which is over 2-fold the number in the Delta variant (7–9). At the time of manuscript submission, the newly emerged Omicron variant had five distinct subvariants (with sublineages): BA.1, BA.2, BA.3, BA.4, and BA.5 (4). An increase in the number of mutations also increases the transmission rate of the virus. The Omicron variant has spread around the globe and produced an immune escape, posing a higher risk of reinfection than the Delta variant (5, 6). Therefore, an urgent need exists for the development of a rapid, simple, and accurate assay that distinguishes between the VOCs, particularly Delta and Omicron.

There are three main methods for SARS-CoV-2 detection: traditional culture, immunological assay, and molecular analysis (10–12). On the one hand, the culture method is the gold standard for pathogen identification. Obtaining viral isolates is the basis for the development of detection reagents, vaccines, and therapeutic antibodies. However, this method is technically demanding and time-consuming and is thus not widely used for the early screening of SARS-CoV-2. On the other hand, the immunological detection method is simple and rapid, and the test results can be obtained within 15 min, which circumvents certain limitations of professional technicians and workplaces (11). Immunological methods mainly involve SARS-CoV-2-specific antigen and antibody tests. Antigen testing is based on the direct detection of SARS-CoV-2-specific antibodies; therefore, the detection results provide direct evidence of early SARS-CoV-2 infection. The samples used in antigen tests are commonly the nasopharyngeal and oropharyngeal swabs and bronchoalveolar lavage specimens collected at infection sites. However, test results are greatly affected by specimen quality, sampling sites, and viral load, which are likely to produce false-negative results. Antibody testing is important for dynamically observing the level of specific antibodies in combination with an analysis of the epidemiological information of the patient. Unfortunately, this poses a less sensitive detection method. As a result, the molecular detection of viral nucleic acids remains the gold standard for diagnosing COVID-19.

Currently, molecular detection methods mainly comprise real-time PCR and isothermal amplification assays (13–15), despite the advantages of the nucleic acid assay. Because the number of novel SARS-CoV-2 variants is constantly increasing and spreading rapidly, a low-cost, flexible, simple, and convenient technique for detecting this virus is urgently needed. In alignment with these requirements, we propose an assay based on high-resolution melting (HRM) that can rapidly and consistently identify viral mutations and variant strains and does not require sequencing. HRM is an easy and reliable closed-tube analysis that has been applied for the identification of mutations by scanning or molecular typing in several research fields, such as epidemiology and

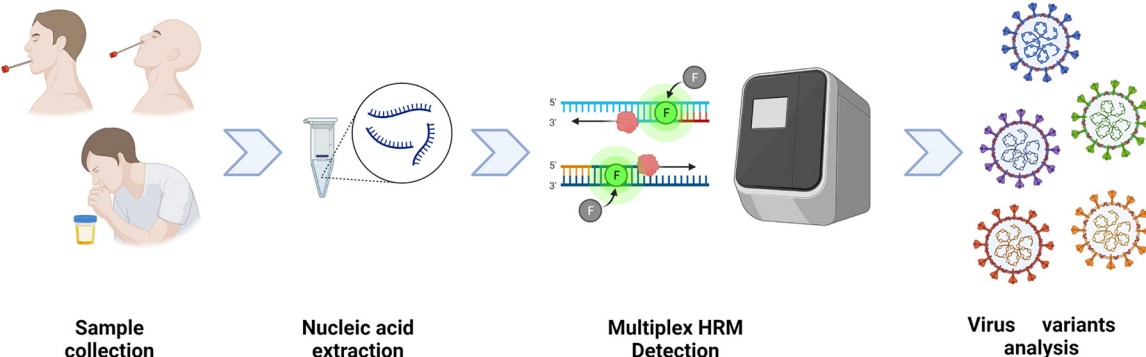

**FIG 1** A flowchart of the multiplex HRM method. The workflow contained four steps. First, respiratory samples were collected. Second, the nasopharyngeal swab and sputum clinical specimens underwent nucleic acid extraction. Third, RNA was added to four assays (assays 1 to 4), each conducted three times for viral variant identification. Samples were regarded as SARS-CoV-2 positive only if the specific melting peaks of the $N$ gene in each assay (the nucleocapsid genes) were observed. Marker mutations were distinguished according to the corresponding $T_m$ value inferred from the melting curves. Lastly, variants were identified based on the profiles of eight marker mutations using the Nextstrain and GISAID databases. SARS-CoV-2 variants are named according to their labels assigned by the World Health Organization (WHO).

microbiology. Exploiting the advantages of HRM, our team has used this approach for the detection of several pathogens and their mutation sites, for example, in *Neisseria gonorrhoeae*, *Chlamydia trachomatis*, *Mycoplasma genitalium*, and SARS-CoV-2 (16–19). In these works, we developed a one-step multiplex HRM method for SARS-CoV-2 detection and mutation site identification, consisting of five assays: one assay detected SARS-CoV-2, and the other four assays obtained 12 mutation sites for variant analysis (19). Moreover, this method proved to be rapid, inexpensive, and flexible, showing high sensitivity and accuracy, and exhibited no cross-reactivity with other respiratory pathogens. Owing to the genomic characterization of SARS-CoV-2 as an RNA virus, it has very high mutation frequencies. Therefore, based on our original work, the current study developed an upgraded multiplex HRM detection method for the analysis of key SARS-CoV-2 mutation sites. The upgraded method presented here shows improvements mainly in two regards. First, because of the flexibility of this technique, significant new mutation sites were identified and introduced into a panel that can be used to pinpoint and analyze SARS-CoV-2 VOCs, such as Delta and Omicron. Second, we increased the reference gene (the conserved sequence of the $N$ gene, which encodes the structural coronavirus nucleocapsid protein) in each assay as a tool for calibrating the amplicon theoretical melting temperature ($T_m$) measurements, thereby reducing the normal error of operation and guaranteeing the consistency of results across different batches. Therefore, our upgraded multiplex HRM detection method enables the rapid screening of SARS-CoV-2 variants to allow the implementation of targeted prevention and control measures for effective monitoring of viral transmission.

## RESULTS

**Developing and optimizing the multiplex HRM assay.** We developed and assessed a multiplex HRM method for the detection of SARS-CoV-2 variant strains, consisting of four triplex assays (assays 1 to 4). To increase the specificity and accuracy of SARS-CoV-2 detection, each assay added a conserved sequence of the $N$ gene of SARS-CoV-2 as an interpretation standard. To provide more detailed variant strain information, the four triple assays were used to analyze eight marker mutations (according to the latest nomenclature system for VOC variants introduced by the WHO): R158G, G142D, Y505H, V213G, S413R, G446S, F486V, and S704L. The experimental workflow is illustrated in Fig. 1. The optimized concentrations of primers allowed consistent heights of the curve peaks. Table 1 lists the final primer sequences, volumes added to each assay, and target genes. As shown in Table 2, R158G and S704L point mutations occurred in the Delta variant. The BA.1 Omicron variant was identified with G142D, Y505H, G446S, and S704L point mutations, while BA.2 Omicron

**TABLE 1** Details of the four assays used in the multiplex HRM method

| Assay | Target | Primer | Sequence (5′–3′) | Vol (μL)[a] |
|---|---|---|---|---|
| Assay 1 | N | N-F | AGTCACACCTTCGGGAACG | 1 |
| | | N-R | AGGCCTGAGTTGAGTCAGCA | 1 |
| | S | G142D-F | CTGTGAATTTCAATTTTGTAATGATCCATT | 0.5 |
| | | G142D-R | CAACTTTTGTTGTTTTTGTGGTAATAAACA | 0.5 |
| | S | R158G-F | AAAGTTGGATGGAAAGTGAGTTC | 1 |
| | | R158G-R | AGGCTGAGAGACATATTCAAAAGTG | 1 |
| | | | | |
| Assay 2 | N | | | 1 |
| | | | | 1 |
| | S | Y505H-F | TCCAACCCACTAATGGTGTTG | 1 |
| | | Y505H-R | AGAAAGTACTACTACTCTGTATGGT | 1 |
| | S | V213G-F | CTAAGCACACGCCTATTAATTTAG | 1 |
| | | V213G-R | CCTGAGGGAGATCACGC | 1 |
| | | | | |
| Assay 3 | N | | | 1 |
| | | | | 1 |
| | S | G446S-F | ATTCTAACAATCTTGATTCTAAGGTT | 0.5 |
| | | G446S-R | CAATCTATACAGGTAATTATAATTACCAC | 0.5 |
| | N | S413R-F | CTGCTGCAGATTTGGATGATT | 0.5 |
| | | S413R-R | TTAGGCCTGAGTTGAGTCAG | 0.5 |
| | | | | |
| Assay 4 | N | | | 1 |
| | | | | 1 |
| | S | F486V-F | ATCTTGATTCTAAGGTTGGTGGTAATTATA | 0.5 |
| | | F486V-R | GGTTGGAAACCATATGATTGTAAAGGAAA | 0.5 |
| | S | S704L-F | GGCGGGCACGTAGTGTAG | 1 |
| | | S704L-R | TAGACACTGGTAGAATTTCTGTGGTA | 1 |

[a]Volume of each 10 μM primer added to the primer pool.

was identified with the mutation sites of G142D, Y505H, V213G, S413R, and S704L, and the BA.5 Omicron variant with those of G142D, Y505H, G446S, S413R, F486V, and S704L.

**Establishing the interpretation criteria.** The $T_m$ ranges of amplicons were established across four assays designed with viral isolates, each performed 10 times with the multiplex method shown in Table 2; these values served as interpretation criteria. Based on the $T_m$ value obtained via the one-step multiplex HRM method and the resultant creation of standards (Table 2), we could determine whether a sample was SARS-CoV-2 positive, establish the base of marker sites, and identify the viral strain variant.

**Analytical performance of the method.** Each dilution of standards (viral isolates mixed in equal proportions with the human genome) was tested 11 times, and the limit

**TABLE 2** Interpretation criteria for results of the multiplex HRM methods (assays 1 to 4)

| Assay | Mutation site | $T_m$ (°C) value range (95% CI[a]), base, for SARS-CoV-2 variant strain: | | | |
|---|---|---|---|---|---|
| | | Delta | Omicron-BA.1 | Omicron-BA.2 | Omicron-BA.5 |
| 1 | N | 83.1–83.9 | 83.1–83.9 | 83.1–83.9 | 83.1–83.9 |
| | R158G | 74.8–75.0, G | 74.3–74.4, A | 74.3–74.4, A | 74.3–74.4, A |
| | G142D | 79.8–79.9, G | 79.3–79.4, A | 79.3–79.4, A | 79.3–79.4, A |
| | | | | | |
| 2 | N | 83.1–83.9 | 83.1–83.9 | 83.1–83.9 | 83.1–83.9 |
| | Y505H | 79.1–79.2, T | 79.6–79.8, G | 79.6–79.8, G | 79.6–79.8, G |
| | V213G | 74.5–74.6, T | 74.5–74.6, T | 75.1–75.2, G | 74.5–74.6, T |
| | | | | | |
| 3 | N | 83.1–83.9 | 83.1–83.9 | 83.1–83.9 | 83.1–83.9 |
| | S413R | 73.8–73.9, A | 73.8–73.9, A | 74.6–74.9, G | 74.6–74.9, G |
| | G446S | 78.8–78.9, G | 78.1–78.2, T | 78.8–78.9, G | 78.1–78.2, T |
| | | | | | |
| 4 | N | 83.1–83.9 | 83.1–83.9 | 83.1–83.9 | 83.1–83.9 |
| | F486V | 74.0–74.2, T | 74.0–74.2, T | 74.0–74.2, T | 74.8–80.0, G |
| | S704L | 78.8–79.0, C | 78.8–79.0, C | 78.8–79.0, C | 78.8–79.0, C |

[a]CI, confidence interval.

**TABLE 3** LOD of each target calculated using regression probit analysis

| Target | No. of positive results/no. of replicates (%) for each no. of dilution copies/reactions | | | | | | | | LOD,[a] copies/reaction (95% CI[b]) |
|---|---|---|---|---|---|---|---|---|---|
| | 10,000 | 1,000 | 100 | 50 | 20 | 10 | 5 | 1 | |
| N | 11/11 (100) | 11/11 (100) | 11/11 (100) | 11/11 (100) | 11/11 (100) | 11/11 (100) | 9/11 (81.8) | 5/11 (45.5) | 7.38 (4.73–35.16) |
| G142D | 11/11 (100) | 11/11 (100) | 11/11 (100) | 11/11 (100) | 11/11 (100) | 11/11 (100) | 6/11 (54.5) | 4/11 (36.4) | 9.72 (6.90–22.12) |
| R158G | 11/11 (100) | 11/11 (100) | 11/11 (100) | 11/11 (100) | 11/11 (100) | 11/11 (100) | 5/11 (45.5) | 3/11 (27.3) | 9.96 (7.31–19.54) |
| Y505H | 11/11 (100) | 11/11 (100) | 11/11 (100) | 11/11 (100) | 11/11 (100) | 11/11 (100) | 5/11 (45.5) | 3/11 (27.3) | 9.96 (7.31–19.54) |
| V213G | 11/11 (100) | 11/11 (100) | 11/11 (100) | 11/11 (100) | 11/11 (100) | 11/11 (100) | 5/11 (45.5) | 2/11 (18.2) | 9.50 (7.16–17.1) |
| G446S | 11/11 (100) | 11/11 (100) | 11/11 (100) | 11/11 (100) | 11/11 (100) | 11/11 (100) | 8/11 (72.7) | 3/11 (27.3) | 7.80 (5.58–16.95) |
| S413R | 11/11 (100) | 11/11 (100) | 11/11 (100) | 11/11 (100) | 11/11 (100) | 11/11 (100) | 6/11 (54.5) | 3/11 (27.3) | 9.33 (6.80–18.72) |
| F486V | 11/11 (100) | 11/11 (100) | 11/11 (100) | 11/11 (100) | 11/11 (100) | 11/11 (100) | 10/11 (90.9) | 7/11 (63.6) | 8.25 (4.91–34.98) |
| S704L | 11/11 (100) | 11/11 (100) | 11/11 (100) | 11/11 (100) | 11/11 (100) | 11/11 (100) | 10/11 (90.9) | 7/11 (63.6) | 8.25 (4.91–34.98) |

[a]LOD, limit of detection.
[b]CI, confidence interval.

of detection (LOD) was determined using probit regression analysis at a 95% detection level. For each target, the LOD was <10 copies/reaction (Table 3; see also Fig. S1 in the supplemental material). There was also no cross-reaction between the SARS-CoV-2 targets within assays 1 to 4 and a viral/bacterial panel used for specificity testing (Table 4).

**Applying the upgraded multiplex HRM to clinical samples.** RNA from 324 nasopharyngeal swabs and sputum samples collected from patients with pneumonia or suspected SARS-CoV-2 infection was analyzed using reverse transcription-quantitative PCR (qRT-PCR) coupled with Sanger sequencing and the multiplex HRM method. Viral RNA extraction was performed in a biosafety level 3 laboratory. Nucleic acids were extracted using the Qiagen Viral RNA minikit (Qiagen, CA, USA) according to the manufacturer's instructions. For multiplex HRM analysis, samples were regarded as SARS-CoV-2 positive only if specific melting peaks of the *N* gene were observed in assays 1 to 4. The entire HRM reaction and analysis of the curves lasted approximately 2 h (not including RNA extraction). Otherwise, the test was repeated for further confirmation. Following this protocol, 48 out of 324 samples tested positive for SARS-CoV-2 via the multiplex HRM method, which was in concordance with the results of qRT-PCR. The remaining 276 samples were also confirmed to be SARS-CoV-2 negative via both methods.

Next, using the results of our HRM assays and our interpretation criteria (Table 2), we determined the bases of eight marker mutations of the 48 SARS-CoV-2-positive samples and compared our results to those of Sanger sequencing to verify our method's accuracy in pinpointing the viral variant strain. Except for one sample with qRT-PCR cycle threshold values of >35, the bases of marker mutations in the other 47 samples were identical to those determined by Sanger sequencing, producing an overall 97.9% (47/48) rate of agreement between the tests. The results demonstrated that, although our interpretation criteria were based on the $T_m$ values obtained from viral isolates of Delta, Omicron-BA.1, Omicron-BA.2, and Omicron-BA.5 variants, our assay can analyze RNA extracted from clinical specimens efficiently.

Finally, the profiles of the eight marker mutations of 48 SARS-CoV-2-positive samples were investigated. Variant strains were identified based on the latest Global Initiative on Sharing Avian Influenza Data nomenclature (https://www.gisaid.org/references/statements-clarifications/) and the Nextstrain database (https://nextstrain.org/). Among the 48 samples, we identified Delta (*n* = 16), Omicron-BA.1 (*n* = 16), Omicron-BA.2 (*n* = 7), and Omicron-BA.5 (*n* = 8) variants. Figure 2 provides representative HRM curves obtained for four variant strains (Delta, Omicron-BA.1, Omicron-BA.2, and Omicron-BA.5).

All in all, the whole collected clinical samples were tested using qRT-PCR, including 48 SARS-CoV-2-positive samples and 276 SARS-CoV-2-negative samples. The mutation sites of 48 SARS-CoV-2-positive specimens were analyzed by the Sanger sequencing method. The results show that there were 16 Delta, 16 Omicron-BA.1, 7 Omicron-BA.2, and 8 Omicron-BA.5 type variants. One sample could not be tested because some marker mutations could not be identified via multiplex HRM and Sanger sequencing. We speculated that this detection failure resulted from a low viral load in this sample, as

**TABLE 4** Specificity evaluation of multiplex HRM method

| No. | Sample | No. of samples | Result with assay[b]: | | | |
|-----|--------|----------------|---------|---------|---------|---------|
| | | | Assay 1 | Assay 2 | Assay 3 | Assay 4 |
| 1 | Adenovirus | 2 | − | − | − | − |
| 2 | Human enterovirus | 2 | − | − | − | − |
| 3 | Human coronavirus OC43 | 2 | − | − | − | − |
| 4 | Human coronavirus 229E | 2 | − | − | − | − |
| 5 | Human coronavirus NL63 | 2 | − | − | − | − |
| 6 | Human coronavirus HKU1 | 2 | − | − | − | − |
| 7 | Middle East respiratory syndrome CoV | 1 | − | − | − | − |
| 8 | Human bocavirus 1 | 2 | − | − | − | − |
| 9 | Human metapneumovirus A | 2 | − | − | − | − |
| 10 | Human metapneumovirus B | 2 | − | − | − | − |
| 11 | Human rhinovirus | 2 | − | − | − | − |
| 12 | Influenza A virus H1N1 | 3 | − | − | − | − |
| 13 | Influenza A virus H3N2 | 2 | − | − | − | − |
| 14 | Influenza B virus | 2 | − | − | − | − |
| 15 | Parainfluenza virus 1 | 2 | − | − | − | − |
| 16 | Parainfluenza virus 2 | 2 | − | − | − | − |
| 17 | Parainfluenza virus 3 | 2 | − | − | − | − |
| 18 | Parainfluenza virus 4 | 2 | − | − | − | − |
| 19 | Respiratory syncytial virus A | 2 | − | − | − | − |
| 20 | Respiratory syncytial virus B | 2 | − | − | − | − |
| 21 | SARS-like coronavirus[a] | 1 | − | − | − | − |
| 22 | *Legionella pneumophila* | 2 | − | − | − | − |
| 23 | *Bordetella pertussis* | 2 | − | − | − | − |
| 24 | *Mycoplasma pneumoniae* | 2 | − | − | − | − |
| 25 | *Chlamydophila pneumoniae* | 1 | − | − | − | − |
| 26 | *Haemophilus influenzae* | 2 | − | − | − | − |
| 27 | *Staphylococcus aureus* | 2 | − | − | − | − |
| 28 | *Moraxella catarrhalis* | 1 | − | − | − | − |
| 29 | *Klebsiella pneumoniae* | 3 | − | − | − | − |
| 30 | *Pseudomonas aeruginosa* | 2 | − | − | − | − |
| 31 | *Acinetobacter baumannii* | 2 | − | − | − | − |
| 32 | *Streptococcus pneumoniae* | 1 | − | − | − | − |
| 33 | *Escherichia coli* | 2 | − | − | − | − |
| 34 | *Neisseria meningitidis* | 1 | − | − | − | − |

[a]This sample was collected from a bat.
[b]Assays 1 to 4 are the variant detection assays of the multiplex HRM method. +, positive detection; −, negative detection.

the PCR cycle threshold value was >35. Meanwhile, the total 324 specimens were detected with multiplex HRM (the detection results as mentioned above), and the results are consistent with those of qRT-PCR combined with Sanger sequencing, except for one sample.

## DISCUSSION

The Delta (B.1.617.2) variant of SARS-CoV-2 was first detected in India in December 2020, and, within a few months, it was widespread (5, 20–23). Delta VOCs possess almost 40 to 60% higher transmissibility than Alpha VOCs, a phenomenon that is closely related to their increased number of mutation sites. More than 10 mutations are present in the *S* gene of the Delta virus (such as R158G, G142D, L452R, and E484Q), which have been associated with increased infectivity and transmissibility. Following the appearance of Delta variants, the SARS-CoV-2 Omicron variant was identified in South Africa in November 2021. Since then, it has spread rapidly and now accounts for the majority of SARS-CoV-2 infections worldwide. At its receptor binding domain (RBD), the *S* gene of this variant has 15 additional mutation sites compared to the RBD wild strain (such as T19I, G142D, Y505H, V213G, G446S, F486V, and S704L); these mutations are closely associated with not only virus infectivity but also immune escape ability (24). Notably, a detection of the most significant mutation sites of the Omicron variant was possible with our upgraded multiplex HRM assay. We used mutations localized in the *S* gene (V213G, R158G, G142D, and S704L) and *N* gene (S413R) of SARS-CoV-2 to identify and analyze its variants, such as Delta and

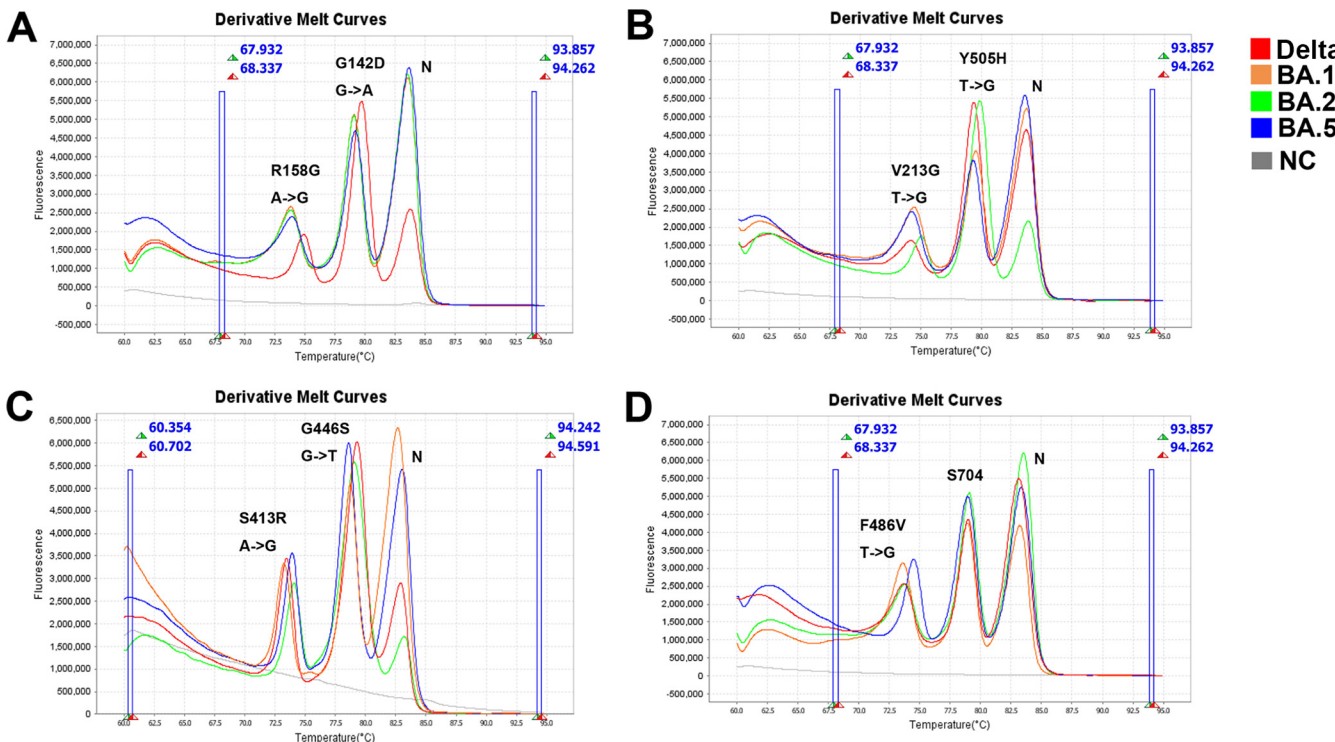

**FIG 2** Melting curve of the multiplex HRM method used for the clinical specimens. Forty-eight SARS-CoV-2-positive clinical samples were detected using the multiplex HRM method. (A) Clinical specimens of the SARS-CoV-2 Delta, Omicron-BA.1, Omicron-BA.2, and Omicron-BA.5 variant strains were analyzed using assay 1. The melting curve showed three independent peaks representative of three targets: nucleocapsid genes and the S413R and G446S mutation sites. (B) The variant samples were similarly tested with assay 2, with peaks representing nucleocapsid genes and the V213G and Y505H mutation sites. (C) The specimens were further investigated using assay 3, revealing melting curve peaks for nucleocapsid genes and the G446S and S413R mutation sites. (D) Lastly, clinical samples were subjected to assay 4, in which melting curve peaks represented nucleocapsid genes and the F486V and S704 mutation sites. S704L can be used to distinguish the SARS-CoV-2 Omicron-BA.2.12.1 variant, which was not included in our collected clinical specimens. Red, orange, green, and blue denote the SARS-CoV-2 Delta, Omicron-BA.1, Omicron-BA.2, and Omicron-BA.5 variants, respectively. Gray denotes the nuclease-free water that was used as a negative control (NC).

Omicron (4, 25). Of particular importance are the mutation sites of G446S, Y505H, and F486V in the RBD region, which are associated with enhanced immune escape ability, increased transmission speed, and reduced antibody response capacity (26–28).

Currently, nucleic acid-based detection methods are the primary means for detecting SARS-CoV-2, either before or after disease onset. Using next-generation sequencing technology, whole-genome sequencing can analyze the entire SARS-CoV-2 genome in detail (29, 30). However, this method poses some practical complications, such as a high running cost and low efficiency in its requirement of samples with a high volume and complete genome sequences. Sanger sequencing has also been used to detect and validate selected SARS-CoV-2 variants. Similarly, this method is time-consuming and involves cumbersome operations. Therefore, a convenient, efficient, and economic method for SARS-CoV-2 detection is urgently needed. This study established an HRM-based assay for the detection of SARS-CoV-2 variants, without the need for genomic sequencing. Our method integrates reverse transcription and multiplex PCR amplification followed by HRM analysis and can facilitate the identification of SARS-CoV-2 variant strains in a one-step, single-tube reaction. This method is useful for the rapid diagnosis of SARS-CoV-2 infection in COVID-19 patients, asymptomatic carriers, and people with suspected infections. In addition, this method could also provide a screening method to improve the surveillance of significant mutations, reducing the viral detection time to approximately 2 h. Some limitations exist in this upgraded HRM-based method. It cannot identify the specific sequence of the unknown or new mutation sites, unlike second-generation sequencing platforms. Moreover, this method still requires thermal cycling equipment, rather than isothermal amplification technology. In addition, our method carries out only triplex detection for each assay.

However, our assay can supplement current SARS-CoV-2 variant detection techniques, offering several advantages. First, in terms of the detection reagent, the multiplex HRM method requires only a single common primer pair for target mutation site testing without the need for a fluorescent probe. Moreover, a number of studies have indicated that EvaGreen dye is sensitive, reliable, and cost-effective (17, 31, 32), which is why EvaGreen was used in this study. Second, with respect to experimental design, the multiplex HRM method confers great flexibility and a low-cost alternative for analyzing viral genomes with high mutation rates. With the increase in the number of viral variant strains, new and significant mutation sites associated with enhanced transmissibility and immune escape have been increasingly superposed. The multiplex HRM method can rapidly analyze new viral variants at key mutation sites, which can prove to be very beneficial for monitoring SARS-CoV-2 transmission and tracing virus variants. Finally, in terms of reliability of the multiplex HRM method, we introduced the conserved sequence of nucleocapsid protein as a $T_m$ value correction standard to guarantee the accuracy and consistency of the detection results across different instruments, laboratories, and batches. In addition, we evaluated the LOD of our method with virus isolates mixed with human genomes, which better mimics actual clinical specimens. Last but not least, our method uses eight important and updated mutation sites (R158G, G142D, Y505H, V213G, S413R, G446S, F486V, and S704L) for the identification of SARS-CoV-2 variants. At the same time, this method is more convenient and faster than sequencing technology. Therefore, owing to the flexibility and scalability of our method, it is a rapid and economical alternative to conventional sequencing-based methods for identifying known mutation sites in SARS-CoV-2.

## MATERIALS AND METHODS

**Primer design and optimization.** Candidate primers for the nine target genes (the nucleocapsid gene plus the eight marker mutations R158G, G142D, Y505H, V213G, S413R, G446S, F486V, and S704L) were designed using Geneious v.11.0.3 software (Biomatters, Auckland, New Zealand), and their specificity was confirmed using the National Center for Biotechnology Information Primer-BLAST tool (http://www.ncbi.nlm.nih.gov/tools/primer-blast/). The $T_m$ values of the amplicons were calculated using OligoCalc (http://biotools.nubic.northwestern.edu/OligoCalc.html). The ideal combinations of primers should fulfill the requirement that the $T_m$ ranges of different amplicons within an assay do not overlap. Precise $T_m$ ranges were obtained via 10 repeats of the experiments with any variant strain (Delta, Omicron-BA.1, Omicron-BA.2, and Omicron-BA.5) using the multiplex HRM method (described below). The lower and upper 95% confidence intervals for all outcomes of each target were calculated and used as interpretation criteria. Accordingly, the concentration of each primer could be adjusted to attain consistency in the height of melting curve peaks.

**Multiplex HRM.** Figure 1 shows the experimental workflow and research design. Specifically, the multiplex HRM reaction required 10 $\mu$L 2× reaction mix (Invitrogen, Carlsbad, CA, USA), 1 $\mu$L 20× EvaGreen fluorescent dye (Biotium, Hayward, CA, USA), 1 $\mu$L SuperScript III RT/Platinum *Taq* mix (Invitrogen), the corresponding volumes of primer pools for each assay (Table 1), 2 $\mu$L of the RNA template, and nuclease-free water, rendering a total reaction volume of 20 $\mu$L. The multiplex HRM reaction conditions were as follows: reverse transcription-PCR for 30 min at 55°C; PCR activation for 2 min at 95°C; 40 cycles of amplification for 30 s at 94°C and 15 s at 53°C and 15 s of signal collection at 68°C; a final HRM step of 95°C for 15 s, 60°C for 1 min, and 95°C for 15 s, and continuous signal collection from 60 to 95°C at an increment rate of 0.025°C/s; and then cooling at 60°C. One-step multiplex HRM was performed using the Applied Biosystems QuantStudio 6 Flex real-time PCR instrument with a fast 96-well block and HRM capacity (Applied Biosystems, Inc., Waltham, MA, USA). The melting curve and $T_m$ values were analyzed using the HRM module in QuantStudio real-time PCR software v1.2.

**Specimens and extraction of viral RNA.** A total of 324 clinical nasopharyngeal swabs and sputum samples were collected from patients with pneumonia or suspected SARS-CoV-2 infection. Four viral isolates were collected from a viral culture of throat swabs. The viral RNA and human genome were extracted using the QIAamp viral RNA minikit (Qiagen, Hilden, Germany) following the manufacturer's instructions.

**LOD testing.** Using gradient-diluted recombinant plasmid containing the target gene as standard, the virus strain was quantified by real-time PCR. The viral isolates were mixed in equal proportions with the human genome to prepare standards at a concentration of 500 copies/$\mu$L, to be used for testing different combinations of the candidate primers. The standards were diluted to concentrations of 0.5, 2.5, 5, 10, 25, 50, 500, 5000, 50 000, and 500 000 copies/$\mu$L in RNase-free water to determine the LOD. Primer sets in each assay were used to simultaneously detect corresponding target genes in the LOD test.

**Specificity testing panel.** To create a specificity panel to test for possible cross-reactivity, we used nucleic acids from isolates and clinical samples that tested positive for common pathogens in respiratory

tract infections. The panel was constructed using 64 nasopharyngeal swabs and comprised adenovirus, enterovirus, coronavirus OC43, coronavirus 229E, coronavirus NL63, coronavirus HKU1, Middle East respiratory syndrome coronavirus (MERS-CoV), bocavirus, metapneumoviruses A and B, rhinovirus, influenza A virus H1N1, influenza A virus H3N2, influenza B virus, parainfluenza viruses 1 to 4 (PIV-1 to -4), respiratory syncytial viruses (RSV A and B), *Legionella pneumophila*, *Bordetella pertussis*, *Mycoplasma pneumoniae*, *Chlamydophila pneumoniae*, *Haemophilus influenzae*, *Staphylococcus aureus*, *Moraxella catarrhalis*, *Klebsiella pneumoniae*, *Pseudomonas aeruginosa*, *Acinetobacter baumannii*, *Escherichia coli*, and one isolate each of *Streptococcus pneumoniae* and *Neisseria meningitidis*.

**Application assessment.** Standard qRT-PCR was used to detect the *ORF1a* and *N* genes of SARS-CoV-2 in clinical samples, according to the manufacturer's protocol (Berger Medical Technology, Shanghai, China). Based on the qRT-PCR results, cDNA of SARS-CoV-2-positive samples was synthesized using the SuperScript IV first-strand synthesis system (Invitrogen). This cDNA was used to amplify fragments in remaining SARS-CoV-2-positive samples containing 12 marker sites via nested PCR with the FastStart High Fidelity PCR system (Roche, Basel, Switzerland); the bases of the 12 marker sites were confirmed using Sanger sequencing. All primers used in the nested PCR are listed in Table S1 in the supplemental material.

**Ethical approval.** All experiments were performed according to the ethical standards of the China National Research Committee and were approved by the Institutional Review Board of the Institute of Pathogen Biology, Chinese Academy of Medical Sciences and Peking Union Medical College. All samples were obtained under approved ethical protocols, and informed consent was obtained from each patient.

**Statistical analysis.** LOD was calculated using probit analysis in SPSS v21.0 statistical software (SPSS Inc., Chicago, IL, USA). The $T_m$ range was determined using the 95% confidence interval with GraphPad Prism v7.0 software (GraphPad Software, Inc., San Diego, CA, USA).

## SUPPLEMENTAL MATERIAL

Supplemental material is available online only.

**SUPPLEMENTAL FILE 1**, DOCX file, 0.18 MB.

## ACKNOWLEDGMENTS

This research project is funded by the grants from the CAMS Innovation Fund for Medical Sciences (CIFMS, 2021-I2M-1-038), Non-profit Central Research Institute Fund of Chinese Academy of Medical Sciences (2019PT310029), and the Fundamental Research Funds for the Central Universities (3332021092).

We have declared that no competing financial interests exist.

J.P. contributed to the development of the study design and the coordination of the execution of the study. L.R. and J.P. coordinated the study and reviewed drafts of the manuscript. L.S. drafted the study protocol, analyzed the results, and drafted the manuscript. L.S. and L.W. conducted the experiment. C.Z., Y.X., L.Z., and Z.Z. helped to perform the experiments. All authors read and approved the final version of the paper.

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
