## [Reviewer comments · Microbiology Spectrum]

Microbiology Spectrum

Rapid Detection of Predominant SARS-CoV-2 Variants Using Multiplex High-Resolution Melting Analysis

Liyang Sun, Liqin Wang, Chi Zhang, Yan Xiao, Lulu Zhang, Ziyuan Zhao, Lili Ren, and Junping Peng

Corresponding Author(s): Junping Peng and Lili Ren, Chinese Academy of Medical Sciences & Peking Union Medical College

Review Timeline:

Submission Date:	January 9, 2023
Editorial Decision:	February 21, 2023
Revision Received:	April 18, 2023
Accepted:	April 26, 2023

Editor: Leiliang Zhang

Reviewer(s): The reviewers have opted to remain anonymous.

Transaction Report:

DOI: <https://doi.org/10.1128/spectrum.00055-23>

February 21, 2023

Prof. Junping Peng
Chinese Academy of Medical Sciences & Peking Union Medical College
NHC Key Laboratory of Systems Biology of Pathogens; Key Laboratory of Respiratory Disease Pathogenomics
No.6, Rongjing Eastern Street, BDA
Beijing
China

Re: Spectrum00055-23 (Rapid Detection of Predominant SARS-CoV-2 Variants Using Multiplex High-Resolution Melting Analysis)

Dear Prof. Junping Peng:

Link Not Available

Sincerely,

Leiliang Zhang

Journals Department
Reviewer comments:

Reviewer #1 (Comments for the Author):

This study developed a sequencing-free high-resolution melting (HRM) assay for the direct detection of SARS-CoV-2 variants. This method has good specificity, and the limit of detection was lower than 10 copies/reaction. In terms of variant detection, the results had a 97.9% (47/48) rate of agreement with standard Sanger sequencing. The multiplex HRM assay therefore offers a rapid and simple procedure for detecting SARS-CoV-2 variants. A few minor concerns have to be addressed to increase the rigor of this study. 1) Please describe in detail the results of the 3 methods for testing clinical samples (qRT-PCR via Multiplex HRM via Sanger sequencing) 2) I suggest to merge Table 2 and Table 3 into one Table 3) The LOD calculated in Table 4 is

different from the LOD written in the abstract (10 copies/reaction), please clarify
4) please clarify the HRM analysis is compatible to different qPCR instruments 5) triplex detection is limited, please add in the limitation parts of this study

Reviewer #2 (Comments for the Author):

I have attached a file including comments for the authors.

Reviewer #3 (Comments for the Author):

Overall, the authors did a good job. The article was very well written.

Staff Comments:

Preparing Revision Guidelines

Please return the manuscript within 60 days; if you cannot complete the modification within this time period, please contact me. If you do not wish to modify the manuscript and prefer to submit it to another journal, please notify me of your decision immediately so that the manuscript may be formally withdrawn from consideration by Microbiology Spectrum.

Here are some feedbacks to help the authors improve their work.

Figures are not numbered.

There have been numerous studies exploring different methods for detecting COVID particularly there are papers using HRM for the detection of SARA-Cov-2 variants, author may need to provide a more strong discussion section using those papers to discuss why this paper has novelty over them.

Line 137-140: While the author mentions two improvements of their assay, one of them being the ability to identify new mutation sites, at the same time in the discussion line 259-261 some limitation is that it cannot identify new mutation sites. Line 274-275 is also addressing completely opposite implications.

Line 176-181: The author reported LOD of less than 10 copies per reaction, however, they did not show any results in this regard. Showing melt curves with different copy numbers is recommended.

Reporting the positive predictive value and negative predictive value is highly recommended.

Line 189: The author claims 2 hours for the entire analytical procedure, does that mean that 2 hours is only for HRM and analysis of the curves, or it also includes RNA extraction step? Specifically, clarify the time inclusion.

199-202: The cycle value greater than 35 is used for qPCR, however, in line 313, 30 cycles? Can you claim that HRM is more sensitive in detection compared to qPCR. According to the AB Applied Biosystem it is recommended to use 40 cycles. Author has changed the AB protocol, what was the reason? what if we put the cycle setting to 40, does it make Negative controls to be positive?

Lines 216-217: Rewrite this sentence.

Line 260-262: There are more limitations the author may need to point out here, although there is plenty of paper published on HRM there are some reasons why this method has not been utilized in clinics yet.

Line 323: The author needs to clarify the specimen preparation, Since you are recommending this method for detection you need to specify if you have used any reagent or centrifuge step after sampling for Sputum or nasopharyngeal swabs

Line 329-335: Clarify on how the author prepared a concentration of 500 copies/ μ l using both the human genome and viral isolates.

Tables 2,3 : The T_m value range in tables 2 and 3 is shown with two decimal numbers, however, Figure 2 shows the T_m with one decimal, It is not recommended to report the confidence interval with two decimal numbers while your assay is only able to detect by one decimal.

Figure 2: How do you interpret figure 2 C S413R between different variants, what is your confidence level in differentiating four variants based on these curves? It might be tricky for a lab technician to diagnose this difference by observing that.

Figure 2: capsid N for BA.1 is closer to 82.5 compared to the same variant in other experiments, how author interprets these variations?

Manuscript ID: Spectrum00055-23

Manuscript title: Rapid Detection of Predominant SARS-CoV-2 Variants Using Multiplex High-Resolution Melting Analysis.

Dear editor and reviewers:

Thank you for your comments concerning our manuscript entitled “Rapid Detection of Predominant SARS-CoV-2 Variants Using Multiplex High-Resolution Melting Analysis.” (Manuscript ID: Spectrum00055-23).

We are very grateful for the constructive comments which have significantly improved the manuscript. We revised the relevant part in manuscript according to the reviewers’ advice. The questions were answered below.

Reviewer1

1、 Please describe in detail the results of the 3 methods for testing clinical samples(qRT-PCR via Multiplex HRM via Sanger sequencing)?

Answer:

We sincerely appreciate your constructive suggestion. According to your valuable comments, we have supplemented the detailed description to the “Results” section of the manuscript (Line 217-227).

2、 I suggest to merge Table 2 and Table 3 into one Table?

Answer:

Thank you very much for your suggestion to make our article reasonable and smooth. We have merged table 2 and table 3 into one table, which is added to the text and marked as table 2.

3、 The LOD calculated in Table 4 is different from the LOD written in the abstract (10 copies/reaction), please clarify?

Answer:

Sorry to make you confused. The LOD of each included mutation site is shown in table 4, and the LOD of each site is uniformly described as all less than 10 copies / μ L in the abstract. According to your valuable comments, we added the specific LOD values of each site to the abstract (Line 40-42, highlighted in yellow) to make the article more clearly.

4、 please clarify the HRM analysis is compatible to different qPCR instruments

Answer:

Thank you very much for your valuable advice. HRM has been applied to a variety of devices including Applied Biosystems, ROCH, QIAGEN, and Idaho Technology. Many detection methods were established for the detection and analysis of various pathogens using different instruments [1; 2; 3].

5、 triplex detection is limited, please add in the limitation parts of this study.

Answer:

Thank you very much for your valuable suggestion, which makes our article clearer. We have added the restriction of triplex detection to the discussion section of the article and highlighted (Line 271-272).

References

- [1] M. Perini, A. Piazza, S. Panelli, S. Papaleo, A. Alvaro, F. Vailati, M. Corbella, F. Saluzzo, F. Gona D. Castelli, C. Farina, P. Marone, D.M. Cirillo, A. Cavallero, G.V. Zuccotti, and F. Comandatore, Hypervariable-Locus Melting Typing: a Novel Approach for More Effective High-Resolution Melting-Based Typing, Suitable for Large Microbiological Surveillance Programs. *Microbiol Spectr* 10 (2022) e0100922.
- [2] L. Xiu, L. Wang Y. Li, L. Hu, J. Huang G. Yong, Y. Wang W. Cao, Y. Dong W. Gu, and J. Peng, Multicentre Clinical Evaluation of a Molecular Diagnostic Assay to Identify *Neisseria gonorrhoeae* Infection and Detect Antimicrobial Resistance. *Int J Antimicrob Agents* 61 (2023) 106785.
- [3] F.M. Gazali, M. Nuhamunada R. Nabilla, E. Supriyati, M. S. Hakim, E. Arguni, E.W. Daniwijaya, T. Nuryastuti, S.M. Haryana, T. Wibawa, and N. Wijayanti, Detection of SARS-CoV-2 spike protein D614G mutation by qPCR-HRM analysis. *Heliyon* 7 (2021) e07936.

Reviewer #2 (Comments for the Author):

Here are some feedbacks to help the authors improve their work.

1. Figures are not numbered.

Answer:

I'm very sorry for our carelessness. The number of the picture has been supplemented.

2. There have been numerous studies exploring different methods for detecting COVID particularly there are papers using HRM for the detection of SARA-Cov-2 variants, author may need to provide a more stronger discussion section using those papers to discuss why this paper has novelty over them.

Answer:

Thank you very much for your valuable advice to polish our article. We have added the advantages of this method to the discussion of the article (Line 291-294) .

3. Line 137-140: While the author mentions two improvements of their assay, one of them being the ability to identify new mutation sites, at the same time in the discussion line 259-261 some limitation is that it cannot identify new mutation sites. Line 274-275 is also addressing completely opposite implications.

Answer:

We sincerely appreciate your constructive suggestion. Our method established can identify and detect the suspected new mutation sites, but it can't obtain the specific sequence information of variant like sanger sequencing. Relevant explanations are added to the discussion section of the article and highlighted (Line 268-269) .

4. Line 176-181: The author reported LOD of less than 10 copies per reaction, however, they did not show any results in this regard. Showing melt curves with

different copy numbers is recommended.

Answer:

Thank you very much for your valuable comments and careful checks. We will supplement the results to the supplementary materials.

5. Reporting the positive predictive value and negative predictive value is highly recommended.

Answer:

Thank you for your valuable suggestions. The detection range of T_m value given in the interpretation standard in Table 2 is the positive interpretation value, and the T_m value of wild strains is beyond the range, so as to achieve the purpose of the mutation information identification of the variants.

6. Line 189: The author claims 2 hours for the entire analytical procedure, does that mean that 2 hours is only for HRM and analysis of the curves, or it also includes RNA extraction step? Specifically, clarify the time inclusion.

Answer:

Thank you very much for your valuable suggestion. As described in the results, two hours is the time for HRM reaction and analysis, excluding RNA extraction (Line 191-192).

7. 199-202: The cycle value greater than 35 is used for qPCR, however, in line 313, 30 cycles? Can you claim that HRM is more sensitive in detection compared to qPCR. According to the AB Applied Biosystem it is recommended to use 40 cycles. Author has changed the AB protocol, what was the reason? what if we put the cycle setting to 40 , does it make Negative controls to be positive?

Answer:

Thank you very much for your valuable comments. It has been corrected, and it has been revised in this paper, and it has been revised into 40 cycles (Line 325) .

8. Lines 216-217: Rewrite this sentence.

Answer:

We sincerely appreciate your constructive suggestion. It has been revised according to the constructive comments (Line 217-227).

9. Line 260-262: There are more limitations the author may need to point out here, although there is plenty of paper published on HRM there are some reasons why this method has not been utilized in clinics yet.

Answer:

Thank you very much for your valuable suggestion. HRM has been widely used in many research, but it will take some time for the popularization of HRM-related equipment. With the continuous development of global economy and medical level, we believe the relevant instruments and methods will be widely applied in the detection and identification of clinical samples.

10. Line 323: The author needs to clarify the specimen preparation, since you are recommending this method for detection you need to specify if you have used any reagent or centrifuge step after sampling for Sputum or nasopharyngeal swabs

Answer:

Thank you very much for your valuable advice. According to your opinion, we have added the sample extraction and operation instructions to the results section (Line 184-189) .

11. Line 329-335: Clarify on how the author prepared a concentration of 500 copies/ μ l using both the human genome and viral isolates.

Answer:

Thank you very much for your valuable comments. We have added the specific method of preparation to the results section (Line342-345) .

12. Tables 2, 3 : The T_m value range in tables 2 and 3 is shown with two decimal numbers, however, Figure 2 shows the T_m with one decimal, It is not recommended to report the confidence interval with two decimal numbers while your assay is only able to detect by one decimal.

Answer:

Thank you for your valuable comments. According to your comments, we have changed the T_m value in Table 2 to one decimal (Table 2 and Table 3 are merged into Table 2).

13. Figure 2: How do you interpret figure 2 C S413R between different variants, what is your confidence level in differentiating four variants based on these curves? It might be tricky for a lab technician to diagnose this difference by observing that.

Answer:

Thank you for your valuable advice. The mutant site was analyzed with the T_m value using HRM method instead of only the melting curve, so the mutant strain and wild strain can be objectively identified by the T_m value (table 2).

14. Figure 2: capsid N for BA.1 is closer to 82.5 compared to the same variant in other experiments, how author interprets these variations.

Answer:

Thank you sincerely for your valuable advice. The T_m value of capsid N for BA.1 was 83.3 °C detected with HRM method, which was in the range of T_m value (83.1-83.9 °C). Otherwise, the result of multiplex HRM method was analyzed not only by the melting curve, but also by the T_m value.

Reviewer #3 (Comments for the Author):

Overall, the authors did a good job. The article was very well written.

Answer:

Thank you for your positive comments. It is my great honor to receive your recommendations.

April 26, 2023

Prof. Junping Peng
Chinese Academy of Medical Sciences & Peking Union Medical College
NHC Key Laboratory of Systems Biology of Pathogens; Key Laboratory of Respiratory Disease Pathogenomics
No.6, Rongjing Eastern Street, BDA
Beijing
China

Re: Spectrum00055-23R1 (Rapid Detection of Predominant SARS-CoV-2 Variants Using Multiplex High-Resolution Melting Analysis)

Dear Prof. Junping Peng:

Your manuscript has been accepted, and I am forwarding it to the ASM Journals Department for publication. You will be notified when your proofs are ready to be viewed.

Sincerely,

Leiliang Zhang
Editor, Microbiology Spectrum
